# Prostaglandin Metabolome Profiles in Zebrafish (*Danio rerio*) Exposed to Acetochlor and Butachlor

**DOI:** 10.3390/ijms24043488

**Published:** 2023-02-09

**Authors:** Shenggan Wu, Xinzong Zhou, Weiwei Qin, Xuehua An, Feidi Wang, Lu Lv, Tao Tang, Xinju Liu, Yueping He

**Affiliations:** 1State Key Laboratory for Managing Biotic and Chemical Threats to the Quality and Safety of Agro-Products, Institute of Agro-Product Safety and Nutrition, Zhejiang Academy of Agricultural Sciences, Hangzhou 310021, China; 2Hubei Insect Resources Utilization and Sustainable Pest Management Key Laboratory, College of Plant Science and Technology, Huazhong Agricultural University, Wuhan 430070, China

**Keywords:** prostaglandins, metabolites, isoprostanes, *Danio rerio*, UPLC–ESI–MS/MS-metabolomics

## Abstract

Prostaglandins (PGs) are critically important signaling molecules that play key roles in normal and pathophysiological processes. Many endocrine-disrupting chemicals have been found to suppress PG synthesis; however, studies about the effects of pesticides on PGs are limited. The effects of two known endocrine disrupting herbicides, acetochlor (AC) and butachlor (BC), on PG metabolites in zebrafish (*Danio rerio*) females and males were studied using widely targeted metabolomics analysis based on ultraperformance liquid chromatography—tandem mass spectrometry (UPLC—MS/MS). In total, 40 PG metabolites were detected in 24 zebrafish samples, including female and male samples, with and without exposure to AC or BC at the sub-lethal concentration of 100 μg/L for 96 h. Among them, 19 PGs significantly responded to AC or BC treatment, including 18 PGs that were upregulated. The enzyme-linked immunosorbent assay (ELISA) test in zebrafish showed BC could cause significant upregulation of an isoprostane metabolite, 5-iPF2a-VI, which is positively related to the elevated level of reactive oxygen species (ROS). The present study guides us to conduct a further study to determine whether PG metabolites, including isoprostanes, could be potential biomarkers for chloracetamide herbicides.

## 1. Introduction

Prostaglandins (PGs) are hormone-like chemical messengers that are involved in a diverse range of biological processes in humans and other vertebrates, including immune system regulation and cardiovascular, gastrointestinal, genitourinary, endocrine, respiratory, sexual behaviors, and reproductive processes [1,2]. In addition, PGs have been implicated in a broad array of diseases, such as cancer, inflammation, cardiovascular disease, hypertension, chronic kidney disease, and respiratory diseases, including COVID-19 [1,3,4,5]. Therefore, PGs can be used as biomarkers for a variety of diseases.

Recently, the ecotoxicological risk of emerging pollutants that target the generation of PGs has attracted increasing attention. Human pharmaceuticals such as nonsteroidal anti-inflammatory drugs (NSAIDs), which target the key PG biosynthetic enzyme cyclooxygenase (COX), are universally sold over the counter [6]. For example, several classic NSAIDs like ibuprofen, aspirin, acetaminophen, and indomethacin work by reducing the production of PGs as COX inhibitors. In addition, some known endocrine-disrupting compounds (EDCs) could also interfere with the PG pathway, such as bisphenol A, genistein, diethylstilbestrol, flutamide, and some phthalates [7]. Furthermore, several pesticides were also reported to suppress PG synthesis in vitro (in mouse Sertoli cells), including cypermethrin, cyprodinil, imazalil o-phenylphenol tebuconazole, and linuron [8]. However, in vivo studies to evaluate the effects of pesticides on PG synthesis have not been reported yet. There is a growing body of literature about pesticides that act as EDCs by having toxic effects on human health. Whether these endocrine-disrupting pesticides could interfere with PG synthesis needs to be further assessed.

Chloroacetanilide herbicides are one of the most widely used groups of herbicides. Commonly used chloroacetamide herbicides like acetochlor (AC) and butachlor (BC) were frequently detected in the environment and biota. AC and BC were reported as endocrine disruptors in mammals and vertebrates, resulting in developmental and reproduction toxicity as well as toxic effects on other systems, including cardiotoxicity [9], immunotoxicity [10], neurotoxicity [11], and apoptosis [12]. AC and BC could disrupt the thyroid and sex steroid endocrine systems in the model vertebrate zebrafish (*Danio rerio*) [12,13,14,15,16,17,18,19,20]. In the present study, metabolomic analysis was first applied to investigate the responses of metabolite profiles in zebrafish females and males exposed to AC and BC. Unexpectedly, levels of about a dozen PG metabolites were upregulated in zebrafish after AC and BC exposure. Then enzyme-linked immunosorbent assay (ELISA) assays were conducted to measure the levels of a couple of differential PGs and reactive oxygen species (ROS) in zebrafish females and males exposed to AC and BC.

## 2. Results

### 2.1. Overview of the Metabolomics in Zebrafish Samples

The present study performed a widely targeted metabolomics approach based on the ultraperformance liquid chromatography–electrospray ionization–tandem mass spectrometry (UPLC–ESI–MS) method to analyze differential metabolites in female and male zebrafish samples of four biological replicates with and without AC and BC treatments at the sub-lethal concentration (100 μg/L) for 96 h. A total of 1683 metabolites were detected in these 24 samples from six treatments: AC-treated females (ACF), BC-treated females (BCF), AC-treated males (ACM), BC-treated males (BCM), control females (CKF), and control males (CKM) (Appendix A). The total ion chromatogram (TIC) of the quality control (QC) sample (a mixture of all the samples investigated) and a multipeak detection plot of chemicals in the multiple reaction monitoring (MRM) mode of the same sample are illustrated in Appendix A, implying that our analysis was repeatable and reliable. Pairwise comparisons achieved by orthogonal partial least squares discriminant analysis (OPLS-DA) scatter scores indicated a strong separation between each treatment group and the control group (Appendix A).

### 2.2. Comparison of Metabolomic Profiles

The fold change (FC) value of ≥2 or ≤0.5 and variable importance in projection (VIP) value of ≥1 were combined to screen the differentially expressed metabolites between each treatment group and the control group, which could be visualized through a volcano plot (Figure 1). The number of upregulated metabolites was greatly higher than downregulated metabolites for the pairwise comparisons except for BCM vs. CKM (Figure 1). Specially, 143 metabolites (131 upregulated and 12 downregulated) were significantly regulated by ACF, whereas 201 metabolites (197 upregulated and 14 downregulated), 172 metabolites (142 upregulated and 30 downregulated), and 116 metabolites (54 upregulated and 62 downregulated) were significantly influenced by BCF, ACM, and BCM (Figure 1).

The Kyoto Encyclopedia of Genes and Genomes (KEGG) pathway annotation and enrichment results were performed in the pairwise comparisons (Figure 2). The differentially expressed metabolites in the groups of ACF vs. CKF and BCF vs. CKF were mainly annotated and enriched in arachidonic acid (AA) metabolism, serotonergic synapse, and neuroactive ligand-receptor interactions. The differential metabolites in the ACM vs. CKM group were mainly enriched in AA metabolism, retinol metabolism, vitamin digestion, and absorption. The differential metabolites in the BCM vs. CKM group were mainly mapped into bile secretion. These analyses indicated that AA metabolism in zebrafish was a major differential metabolic pathway responding to AC and BC, besides BCM. It is also of interest that a general sex-dependent pattern of zebrafish metabolomics was observed after exposure to AC and BC.

To analyze the influence of sex on the differential metabolite profiles of zebrafish after being exposed to AC and BC, Venn diagrams showing consistent metabolite profiles between different comparisons were established (Figure 3). A total of 102 common metabolites showed consistent profiles in zebrafish females exposed to AC and BC, most of which (100/102) were upregulated (Figure 3; Appendix A). The number of upregulated eicosanoids (28 members) was higher than that of other metabolite classes, about half of which were PGs (15 members), indicating that both AC and BC could induce the PG metabolite profiles in zebrafish females. However, no eicosanoids existed in the joint different metabolites between ACM vs. CKM and BCM vs. CKM (Figure 3; Appendix A). Only 34 differential metabolites (24 upregulated and 10 downregulated) were found with coincident responses to AC and BC in zebrafish males. Half of the joint different metabolites in male fish exposed to AC and BC were polypeptides. It is interesting to note that zebrafish males exhibit a significantly different response to AC and BC. These results indicated that some key physiological and metabolic activities leading to the synthesis of eicosanoids might be activated under AC and BC in zebrafish, but sex-specific differences persist significantly.

### 2.3. PG Metabolite Profiles in Zebrafish Exposed to AC and BC

In the arachidonic acid (AA) metabolism pathway, AA can be converted into prostaglandins (PGs), thromboxanes, leukotrienes, and lipoxins. PGs are classified based on the structures of their cyclopentane rings (designated as PGA~PGK) and the number of double bonds in their hydrocarbon structures (designated as PGX1, PGX2, and PGX3; “X” indicates “A~K”) (Figure 4). PGX1, PGX2, and PGX3 are biosynthesized through the cyclooxygenase (COX) pathway from different polyunsaturated fatty acids (PUFAs), dihomo-γ-linolenic acid (DGLA), AA, and eicosapentaenoic acid (EPA). Isoprostanes (iPGs) are isomers of conventionally derived PGs and are produced independently of COX enzymes, primarily from the oxidative modification of PUFAs via a free radical-catalyzed mechanism (Figure 4).

In total, 40 PGs (including iPGs) were found in all the detected zebrafish samples (Appendix A). Among them, nine types of PGs (PGA, PGB, PGD, PGE, PGF, PGG, PGI, PGJ, and PGK) were found, including two major PG types (PGFs and PGEs) with 18 and 11 metabolites, respectively (Appendix A; Figure 4). All six series of PGs (PGX1, PGX2, PGX3, iPGX1, iPGX2, and iPGX3) were detected in our zebrafish samples, of which PGX_2_ was the most abundant, with 22 members.

There were 19 PG metabolites with significantly different levels in the female and male samples after AC or BC treatment, with fold change values of ≥2 or ≤0.5 and VIP values of ≥1 (Figure 4; Appendix A). In the ACF treatment, 16 PG metabolites (six PGFs, three PGEs, three iPGXs, two PGAs, and PGD2) showed significant upregulation while one iPG (8-iso PGF_3_) was downregulated. BC treatment caused the upregulation of 18 PGs in female zebrafish, including six PGFs, five PGEs, three iPGXs, three PGAs, and PGD2. In the ACM treatment, nine PG metabolites (three PGFs, three PGAs, two PGEs, and 5-iPF2α-VI) were significantly upregulated. Only two PG metabolites (2,3-dinor-6-keto PGF1α and 8-iso PGF3) in males positively responded to BC treatment.

DGLA, AA, and EPA were also detected in our metabolomic analysis (Figure 4; Appendix A). Only DGLA was upregulated in the ACF group compared to CKF, by 2.3-fold.

### 2.4. Different Levels of PGs in Zebrafish Exposed to AC and BC Using ELISA

ELISA tests were conducted to determine the levels of the total PGs, PGE2, PGD2, 11β-13,14-dihydro-15-keto PGF2α (PGFM), 5-iPF2a-VI, and reactive oxygen species (ROS) (Figure 5). No significant difference was found in the levels of the total PGs, PGE2, PGD2, and PGFM between the pairwise comparisons of the AC/BC and CK treatments. However, BC could cause a significant upregulation of 5-iPF2a-VI whether in female or male zebrafish (*p* < 0.05). Furthermore, ROS levels were significantly increased in the ACF, BCF, and BCM groups (*p* < 0.05). There was a significant positive correlation between the levels of ROS and 5-iPF2a-VI (*F* = 32.4; *p* < 0.0001; *R*^2^ = 0.669).

## 3. Discussion

Previous assays showed that AC and BC could have a range of sub-lethal effects in zebrafish at the phenotype level of individuals and tissues [9,14,15,16,18,19,21], at the transcript level [11,12,13,15,16,17,19,20], and at the hormone level [13,14,15,16,18,20]. The current study used a widely targeted metabolomics approach based on the UPLC–ESI–MS/MS method to examine the sex-specific metabolomic profiles of zebrafish females and males exposed to AC or BC at a sub-lethal concentration of 100 μg/L for 96 h. AC and BC could affect the levels of thyroid hormones, triiodothyronine (T3) and thyroxine (T4), in larvae [13,14,20] and adults [18], as well as steroid hormones (e.g., testosterone and 17β-estradiol) in adults [15,16,18]. However, thyroid hormones and key sex hormones such as testosterone and 17β-estradiol were absent from our metabolomic analysis. Unexpectedly, PGs are one major group of differential metabolites, indicating that AC and BC could affect PG synthesis in zebrafish, which was not reported in previous studies.

There was another report discovering that some pesticides could affect PG synthesis in vitro [8]. Cypermethrin, cyprodinil, imazalil, o-phenylphenol, tebuconazole, and linuron could suppress PGD2 synthesis in mouse Sertoli cells [8]. The supplementation assays with AA and the molecular modeling studies suggested that the sites of action of these pesticides might be COX enzymes [8]. On the contrary, as indicated in our study, AC and BC could upregulate PG levels. The elevated PGs might not be associated with the regulation of their biosynthetic precursors, DGLA, AA, and EPA. Whether AC and BC could target COX enzymes and other PG biosynthetic enzymes remains unknown.

The biosynthetic pathway of PGs is conserved in mammalian species and nonmammalian vertebrates, including fish [2]. Membrane phospholipids are esterified by phospholipase A_2_ to generate free PUFAs. Three PUFAs, DGLA, AA, and EPA, are subsequently converted to PGG1, PGG2, and PGG3 through COX enzymes. PGG is then reduced to PGH by the same enzyme. Then, PGH is metabolized to primary PGs through different PG synthases, which involves a complex orchestration [22]. Until now, the biological functions of only a few core PGs (such as PGE2, PGF2α, PGD2, PGB2, PGI2, and PGE3) and their primary metabolites (such as 15-keto-PGF2α, 13, 14-dihydro-15keto-PGF2α, 15d-PGJ2, and Δ12-PGJ2) have been determined in fish [2,23,24,25,26,27,28,29,30,31]. For example, PGF_2α_ and its primary metabolites (15-keto-PGF2a and 13, 14-dihydro-15-keto-PGF2α) act as reproductive hormones in female fish and are involved in uterine contractions during ovulation and spawning [2,28]. PGF2α is also used as a sex pheromone for regulating sexual behaviors in male fish [27]. PGE2 has been found to be associated with multiple physiological processes, including reproduction [2], immune response and inflammation [25,29], embryonic stem cell development [30], the cardiovascular system [31,32], and the renal system [26]. PGD2 plays a key role in the immune response and inflammation in fish [24,25]. Exposure to AC and BC at sub-lethal doses could cause developmental toxicity in zebrafish embryos [14,16,19,21], induced cardiovascular toxicity and increased locomotor activity in zebrafish larvae [9,21], decreased ovarian development in females [15,18], a downregulated gonadosomatic index in males [15,18], and reduced fecundity rates in zebrafish adults [18]. AC and BC exposures could also affect the transcription patterns of many key genes involved in the hypothalamic-pituitary-gonadal/thyroid (HPG/HPT) axis in zebrafish embryos, larvae, and adults [12,13,15,16,17,19,20]. Furthermore, AC and BC exposures in zebrafish embryos could affect the expression of the cell apoptosis pathway [11,12], innate immunity, as well as oxidative stress [12]. Therefore, a number of upregulated PGs in zebrafish under the exposure to AC and BC observed in our study remain to be further studied regarding their biological functions.

It is undeniable that the upregulated levels of a couple of iPGs could be the consequence of the elevation of ROS. There was good evidence that AC and BC induced oxidative stress in zebrafish [12,15,33,34]. For example, our previous study showed that the relative mRNA levels of oxidative stress-related enzymes (e.g., cat, gpx, and cu/zn-sod) in zebrafish were significantly increased after exposure to 0.19–0.74 μM (50–200 μg/L) of acetochlor for 96 h [12]. Oxidative stress is important in determining the toxicity of xenobiotics, including pesticides [35]. Biomarkers of oxidative stress like hydrogen peroxide (H_2_O_2_), reduced glutathione (GSH), and malondialdehyde (MDA), 8-OH-deoxyguanosine (8-OhdG), isoprostanes, and antioxidant enzymes (e.g., CAT, SOD, GST, and GPX) have been demonstrated to be sensitive to pesticide exposure [35]. There was a significant positive correlation between the levels of ROS and 5-iPF2a-VI in zebrafish, which supports the speculation that the upregulation of iPGs caused by AC and BC could be the consequence of elevated ROS levels.

The present study indicated that iPGs could be novel biomarker candidates for assessing chloroacetanilide herbicides. Several molecular epidemiological studies had provided us with evidence that iPGs showed a significant positive correlation to pesticide exposure [36,37,38]. For example, the level of 8-iPGF2a was significantly related to urinary organophosphorus metabolites in South Korean male farmers [36], to urinary 2,4-Dichlorophenoxyacetic acid (2,4-D) in pesticide applicators in Kansas, USA [37], and to urinary glyphosate in agricultural workers in Brazil [38]. Our metabolomic data showed the 8-iPGF2a level in zebrafish was not affected by AC and BC exposure; however, three iPGs, 8-iso PGF1α, 5-iPF2α-VI, and 8-iso-16-cyclohexyl-tetranor PGE2, were induced by AC and BC. Our ELISA assay showed elevated 5-iPF2α-VI was positively associated with increasing ROS levels. Therefore, 5-iPF2α-VI could be a new oxidative stress biomarker for these herbicides. The molecule 5-iPF2α-VI is an isoprostane from the unique Type VI class of isoprostanes. The molecule 5-iPF2α-VI was reported to be a better predictor of oxidative stress in preeclampsia during pregnancy than 8-iso-PGF2, indicating its possible antioxidative role [39]. However, the relationship between 5-iPF2α-VI and oxidative stress caused by pesticides needs to be further investigated.

## 4. Materials and Methods

### 4.1. Animals and Herbicide Exposure

Adult zebrafish (*D. rerio*) (AB strain and 3 months old) were provided by the Institute of Hydrobiology, Chinese Academy of Sciences (Wuhan, China). The fish were kept at a temperature of 25 ± 1 °C under a photoperiod of 14 h of light and 10 h of darkness. The fish were fed daily with paramecia (*Paramecium caudatum*) and brine shrimp (*Artemia franciscana*).

Acetochlor (AC, technical grade AI: 95%) and butachlor (BC, technical grade AI: 94%) were provided by Nantong Jiangshan Agrochemical and Chemicals Co., Ltd., Nantong, China. AC and BC were dissolved in acetone. The stock solutions (100 mg/L) were stored in the dark at 4 °C. Eight three-month-old females or males were exposed to AC or BC at a sub-lethal concentration of 100 μg/L [12,18] in a 5 L glass tank for 96 h. The 0.1% acetone solution was used as the solvent control. Four biological replications were conducted. Whole body samples were quickly frozen on dry ice and preserved at −80 °C.

### 4.2. Widely Targeted Metabolomics Analysis

The thawed samples, which were 50 ± 2 mg each and mixed with eight females or males per repeat, were crushed with cold steel balls and homogenized at 30 Hz for 3 min. Four repeat samples were used for each treatment. Next, each sample was dissolved in 1 mL 70% methanol containing 0.1 mg L^−1^ of the internal standard 4-Fluoro-L-α-phenylglycine (TCI, Co., Ltd., Taipei, Taiwan) and vortexed vigorously for 5 min. Then, 400 μL of supernatant was collected after centrifugation at 12,000 rpm at 4 °C for 10 min and stored overnight in a refrigerator at −20 °C. The next day, following centrifugation at 12,000 rpm at 4 °C for 3 min, 200 μL of supernatant was collected in the insert of the corresponding injection bottle for on-board analysis. Metabolomics analysis was performed by Wuhan Metware Biotechnology Co., Ltd. (Wuhan, China).

The sample extracts were analyzed using a UPLC–ESI–MS/MS system (UPLC, Exion LCAD, https://sciex.com.cn/, accessed on 22 June 2021). UPLC separations were performed using a Waters ACQUITY UPLC HSS T3 C18 column (1.8 μm, 2.1 mm × 100 mm; Waters Corporation, Milford, CT, USA) that was maintained at 40 °C with mobile phase A set at 0.1% formic acid in ultrapure water and phase B set at 0.1% formic acid in acetonitrile. The gradient program was set to 95:5 *v/v* at 0 min, 10:90 *v/v* at 10.0 min, 10:90 *v/v* at 11.0 min, 95:5 *v/v* at 11.1 min, and 95:5 *v/v* at 14.0 min. The injection volume was set to 5 μL, and the flow rate was 0.4 mL/min.

Each sample extract was used for the widely targeted metabolomics analysis based on the ESI–QTRAP–MS/MS approach. Triple quadrupole scans with linear ion traps were acquired using a triple quadrupole-linear ion trap mass spectrometer system (QTRAP^®^, https://sciex.com/, accessed on 22 June 2021), in which an ESI Turbo Ion-Spray interface was used, the positive and negative ion modes were applied, and Analyst 1.6.3 software (Sciex, Framingham, MA, USA) was controlled. The ESI source conditions were as follows: the gas temperature was 500 °C; the ion source gas 1, ion source gas 2, and curtain gas were 50 Psi, 50 Psi, and 25 Psi, respectively; the ion spray voltage was 5500 V or −4500 V in positive or negative modes, respectively; and the setting parameter of collision-activated dissociation (CAD) was high. Both 10 and 100 μmol/L polypropylene glycol solutions were used in triple quadrupole scans and linear ion trap modes, respectively, for instrument tuning and mass calibration. A specific set of multiple reaction monitoring (MRM) transitions was monitored for each period according to the metabolites eluted during this period.

Raw data from metabolomics analysis were converted to formatted files using Proteo Wizard (Version 3.0.19095-938eda31a, http://proteowizard.sourceforge.net, accessed on 22 June 2021) [40]. Metabolite identification was performed using two annotation approaches. First, an in-house metabolite library (Wuhan Metware Biotechnology Co., Ltd., Wuhan, China) containing a large number of chemical standards was used. Second, metabolites were identified based on accurate mass, isotope pattern, and MS/MS spectra against public metabolite databases, including HMDB (http://www.hmdb.ca/, accessed on June 2018), MoNA (http://mona.fiehnlab.ucdavis.edu/, accessed on April 2021), MassBank (http://www.massbank.jp/, accessed on January 2020), METLIN (https://metlin.scripps.edu, accessed on January 2020), and NIST (https://www.nist.gov/, accessed on April 2021). The MS/MS spectra similarity score was calculated using the forward dot-product algorithm, which considered both fragments and intensities [41]. The similarity cutoff score was set at 0.5.

Metabolite quantification was carried out using an MRM method based on raw data acquired from widely targeted metabolomics. The metabolite abundances in the different samples were quantified according to their MS peak areas. Pairwise comparisons were carried out by orthogonal partial least squares discriminant analysis (OPLS-DA). Metabolites with significant content differences were set with thresholds of variable importance in projection (VIP) ≥1 and fold change ≥2 or ≤0.5. Statistical analyses of the relative content data were performed using SPSS 26.0 software (IBM Corp., Armonk, NY, USA).

### 4.3. Measurement of PGs and ROS by Enzyme-Linked Immunoassay

The samples, mixed with eight females or males per repeat, were crushed and homogenized in the Phosphate Buffered Saline (PBS) solution (0.01 M, pH 7.4). Three repeat samples were used for each treatment. The homogenate was centrifuged at 5000 rpm/min for 10 min and the supernatant was separated. The collected supernatant was then stored at −80 °C. The levels of the total PGs, PGE2, PGD2, 11β-13,14-dihydro-15-keto PGF2α (PGFM), 5-iPF2a-VI, and ROS were detected by commercial ELISA kits (Shanghai Hengyuan Biological Technology Co., Ltd., Shanghai, China) according to the manufacturer’s instructions. These kits employed HRP-conjugate mouse antibodies to quantify PGs and ROS levels in samples. Statistical differences were subjected to one-way analysis of variance (ANOVA), followed by Tukey’s multiple comparison test. The statistical significance level was set at *p* < 0.05.

## 5. Conclusions

The current study provided metabolome-level toxicological information on two chloroacetanilide herbicides in zebrafish. Our results revealed that nineteen PG metabolites, including four iPGs, were significantly regulated by AC and BC in zebrafish adults. Our ELISA assay revealed that an upregulated 5-iPF2-VI level was positively associated with increasing ROS levels. PGs are key hormone-like substances that affect a range of biological functions, including immune, cardiovascular, endocrine, and reproductive processes. Isoprostanes (iPGs) are gold biomarkers for assessing oxidative stress in animals. Previous studies uncovered that chloroacetanilide herbicides could have developmental and reproduction toxicity, cardiotoxicity, immunotoxicity, oxidative stress, and apoptosis in fish. Therefore, our findings signal a need for further studies to investigate whether the regulation of PG levels in zebrafish could be related to these toxic endpoints of AC and BC and whether PGs (especially iPGs) could be novel biomarkers for these chloroacetanilide pesticides.

## Figures and Tables

**Figure 1 ijms-24-03488-f001:**
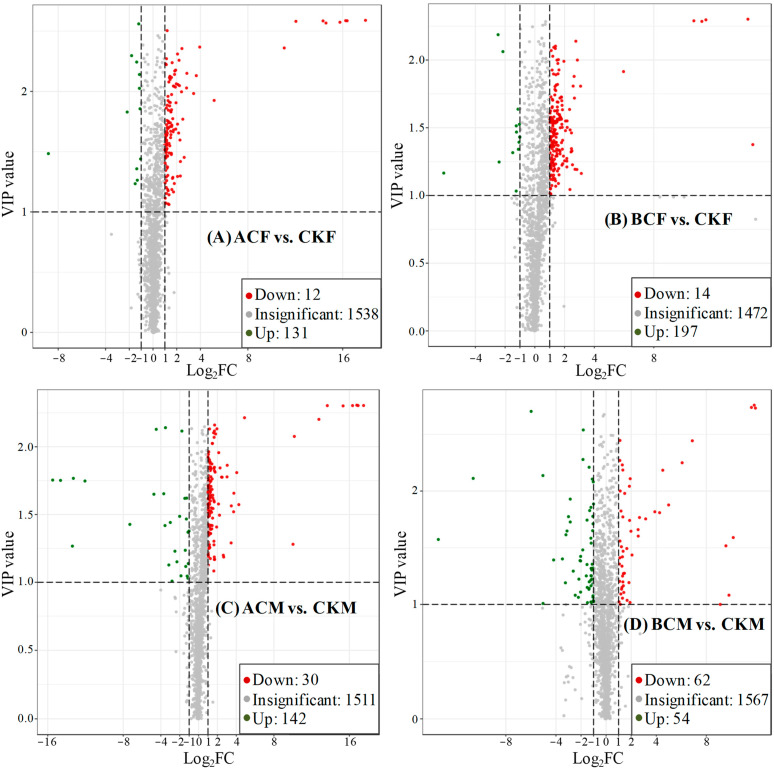
Volcano plots of the differential metabolites of (**A**) ACF vs. CKF; (**B**) BCF vs. CKF; (**C**) ACM vs. CKM; (**D**) BCM vs. CKM. VIP denotes the variable’s importance in the projection value. FC denotes the fold variable value.

**Figure 2 ijms-24-03488-f002:**
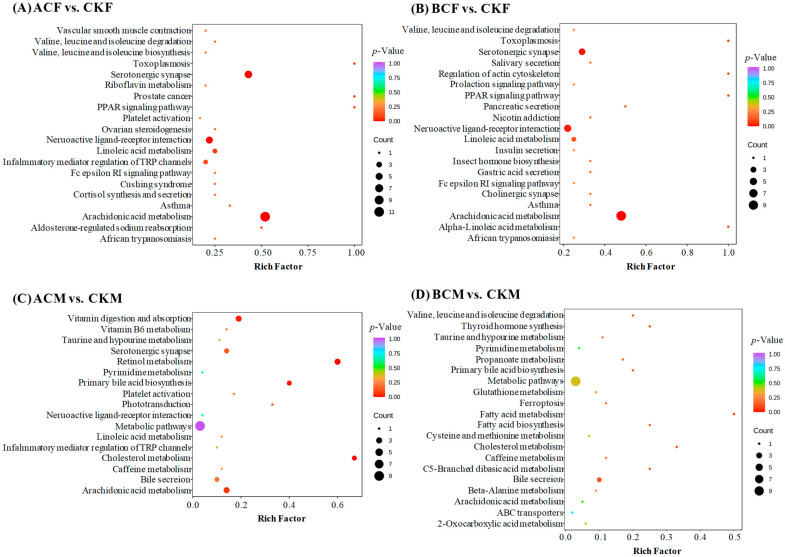
The enriched KEGG pathway terms covered by differential metabolites: (**A**) ACF vs. CKF; (**B**) BCF vs. CKF; (**C**) ACM vs. CKM; (**D**) BCM vs. CKM.

**Figure 3 ijms-24-03488-f003:**
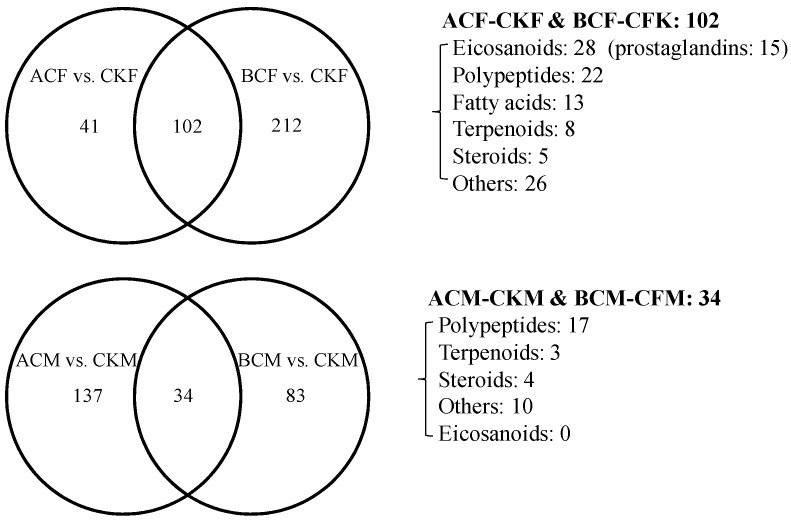
Venn diagrams of the differential metabolites between pairwise comparisons.

**Figure 4 ijms-24-03488-f004:**
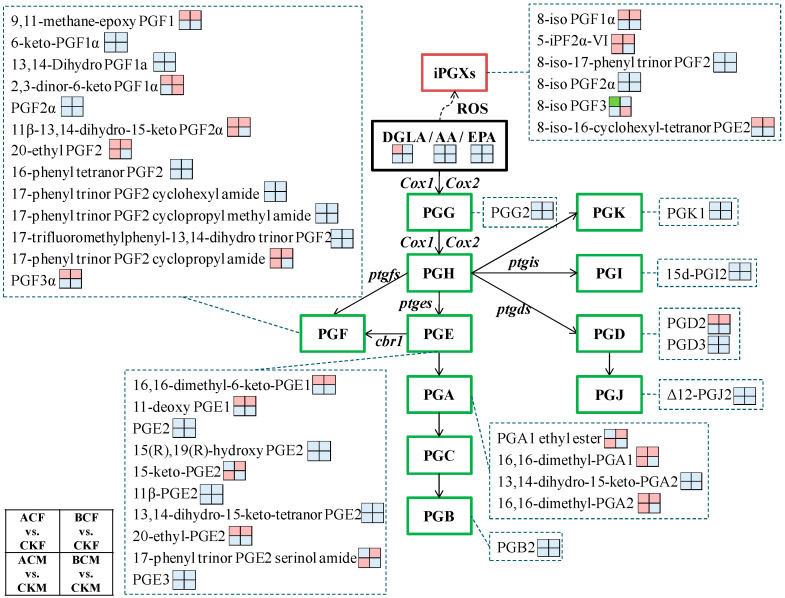
Metabolite profiling of PGs in zebrafish exposed to AC and BC.

**Figure 5 ijms-24-03488-f005:**
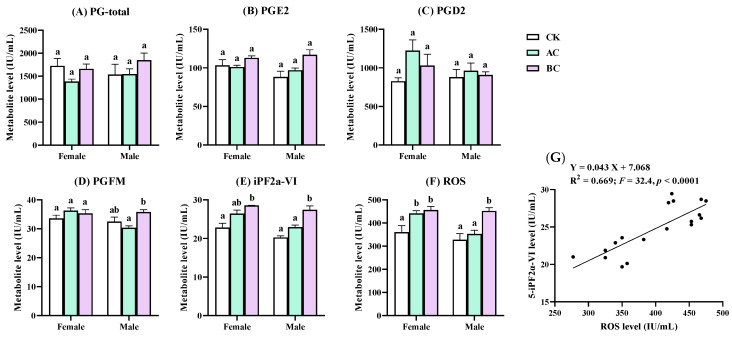
Levels of PGs and ROS determined by ELISA. (**A**–**F**) exhibited the levels of total PG, PGE2, PGD2, PGFM (11β-13,14-dihydro-15-keto PGF2α), 5-iPF2a-VI, and ROS; (**G**) showed a significantly positive correlation between the levels of 5-iPF2a-VI and ROS. Different letters above histograms indicate statistically significant differences among different treatments (*p* < 0.05).

## Data Availability

The authors confirm that the data supporting the findings of this study are available within the article and its Appendix A.

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
