# Peer review of "Prostaglandin Metabolome Profiles in Zebrafish (Danio rerio) Exposed to Acetochlor and Butachlor"

_ijms, 2023, doi:10.3390/ijms24043488_

Round 1

Reviewer 1 Report

The manuscript entitled ‘Prostaglandin Metabolome Profiles in Zebrafish (Danio rerio)  Exposed to Acetochlor and Butachlor’ addresses the study of prostaglandin metabolome using the widely targeted metabolomics approach based on the UPLC–ESI–MS analysis in male and female zebrafish samoles exposed to sublethal doses of herbocide/endocrine disuptor compounds, acetochlor and butachlor. 

The manuscript is scientifically sound and represents significant contribution in the field of ecotoxicology.

All sections of the manuscript followed the scientific mrethodology and were adequaltely structured. 

he introductory part was concise and described the current studies in the field, and the main objectives of the study. The study results are clearly presented in graphical and tabulated form. References were properly coted and adequately corroborated the results of the study.

In my opinion this manuscript is acceptable for the publication after minor spelling changes.

Author Response

In my opinion this manuscript is acceptable for the publication after minor spelling changes.

Response 1: Thank you very much for your comments and suggestions. We checked the spelling errors throughout the text.

Reviewer 2 Report

This is the revision of the manuscript with the title " Prostaglandin Metabolome Profiles in Zebrafish (Danio rerio) Exposed to Acetochlor and Butachlor" by Shenggan Wu & colleagues. In general, the concept of the manuscript is interesting. The text is well written and organized. The work is appropriate for the journal, and the results are interesting. However, there are some observations that must be addressed before the manuscript can be accepted.

L82: the abbreviation must be explained at first use

Fig.1, 2 the quality of the figures needs to be improved.

L269: what "standard internal extract" does mean? Please add details to it.

L273: change liner to insert

L292: what "collision gas was high"  does mean?

L292: change ten to 10

Author Response

Point 1: This is the revision of the manuscript with the title " Prostaglandin Metabolome Profiles in Zebrafish (Danio rerio) Exposed to Acetochlor and Butachlor" by Shenggan Wu & colleagues. In general, the concept of the manuscript is interesting. The text is well written and organized. The work is appropriate for the journal, and the results are interesting. However, there are some observations that must be addressed before the manuscript can be accepted.

Response 1: Great appreciation for your comments and suggestions. The manuscript was improved based on our revisions according to your suggestions.

Point 2: L82: the abbreviation must be explained at first use

Response 2: After checking throughtout the text, a couple of abbreviations were suppled with the explains at first use, such as ELISA, VIP.

Point 3: Fig.1, 2 the quality of the figures needs to be improved.

Response 3: According to your suggestion, Fig.1 and Fig.2 were improved.

Point 4: L269: what "standard internal extract" does mean? Please add details to it.

Response 4: In the widely targeted metabolomics analysis, our samples were dissovlved in 1 mL 70% methanol containing 0.1 mg L-1 internal standard 4-Fluoro-L-α-phenylglycine. This detail was added in the manuscript.

Point 5: L273: change liner to insert

Response 5: Changed.

Point 6: L292: what "collision gas was high"  does mean?

Response 6: Sorry for the confusing description. Actually, the setting parameter of collision-activated dissociation (CAD) was high. This sentence was changed.

Point 7: L292: change ten to 10

Response 7: Changed.